# Influence of Nanoclay on the Thermo-Mechanical Properties of Silicone Pressure-Sensitive Adhesives

**DOI:** 10.3390/ma15217460

**Published:** 2022-10-24

**Authors:** Adrian Krzysztof Antosik, Karolina Mozelewska

**Affiliations:** Department of Chemical Organic Technology and Polymeric Materials, Faculty of Chemical Technology and Engineering, West Pomeranian University of Technology in Szczecin, Piastow Ave. 42, 71-065 Szczecin, Poland

**Keywords:** pressure-sensitive adhesives, thermal crosslinking, silicone additives, dellite

## Abstract

This research was carried on newly obtained innovative materials—self-adhesive one-sided tapes based on silicone pressure-sensitive adhesives. In order to obtain tapes, the stable adhesive composition was subjected to physical modification by incorporating into it various amounts of selected silicon fillers. The produced pressure-sensitive adhesives were tested for viscosity and thermogravimetric analysis, as well as the manufactured tapes; i.e., peel adhesion, tack, cohesion at room and elevated temperature, SAFT test (shear adhesive failure temperature), and shrinkage. The prepared self-adhesive tapes retained their self-adhesive properties at a level close to the initial level while increasing the thermal resistance by 70–75 °C, reaching the level of 220–225 °C. The new self-adhesive materials have application potential and can be used as a material for special applications in the field of electrical engineering and heavy industry.

## 1. Introduction

Clay minerals are minerals made of hydrated aluminum aluminosilicates (as well as magnesium and iron) with a characteristic, layered crystal structure. One of the layers contains silicate tetrahedra, the centers of which are Si^4+^ surrounded by four oxygen atoms. These tetrahedra relate to each other to form a hexagonal system of Si_2_O^2−^_5_ units.

The second layer is formed by aluminum hydroxide octahedrons in which Al^3+^ are surrounded by six hydroxyl groups. The layers connect with each other in characteristic systems that determine their properties and constitute the basis for the classification of clay minerals. These materials are the main components of clay rocks: clay, shale, decay, and soil. They are very common and can also be a part of such sedimentary rocks as marls, mudstones, sandstones, and limestones. Clay minerals include minerals, i.e., the group of kaolinite and serpentine (kaolinite, cover, dickite, halloysite, metahalloysite, antigorite, lizardite, chrysotile, garnierite, nepouite, greenalite, and serpentine), talc and pyrophyllite groups (talc, kerlite), pyrophyllite, montmorillonites (montmorillonite, beidellite, nontronite, and saponite), mica group and hydromikimics (illity, brammalite, glauconite, seladonite, muscovite, phlogopite, paragonite, lepidolite, hydromuscovite pholidoid, hydrobiotchlorite, hydrophytchlorite group, chloramtchlorite group, and group) nimite (pennantite), vermiculite, sepiolite group and palygorskite, imogolite, allophan “group”, and mixed-pack minerals (rectorite, tosudite, cornsite, hydrobiotite, alietite, and kulkeite) [1,2,3].

This fine-grained geologic material exhibits properties of plasticity when wet, but becomes hard, brittle, and non-plastic upon drying or firing. Clay materials usually crystallize in a monoclinic or triclinic crystallographic system. They are characterized by low hardness (up to 2.5 on the Mohs scale). They are a product of weathering transformations taking place on land and seas, as well as-low temperature hydrothermal processes [2,3,4,5].

Because of their properties and application possibilities, they are interesting for scientists. Many of them are used as ceramic raw materials. They are used in the production of insulation and refractory materials, paper, rubber, cosmetics, pharmaceuticals, fertilizers, and dyes. They are used in the electronics, chemical, construction, textile, sugar, and brewing industries. They are used to filter drinking water and are a component of drilling fluid. They rehabilitate areas contaminated with heavy metals and petroleum products. Clay materials are added to polymer matrices to improve the mechanical properties and thermal resistance [3,4,5,6].

Silicone pressure-sensitive adhesives (Si-PSAs) are a relatively new area in the adhesive materials industry. The first products of this type were introduced in the 1960s where they found industrial applications, but it is only since 2000 that the interest in Si-PSA has increased owing to new possible applications such as medical plasters and mounting tapes. Silicone pressure-sensitive adhesives are characterized by high adhesiveness to surfaces with both high and low surface energy. In addition, they are characterized by excellent chemical resistance, excellent resistance to environmental conditions, and very high durability. Thanks to this, Si-PSAs have a very high resistance to aging. These adhesives exhibit relatively high thermal resistance (compared with their organic counterparts), thus they can be used as a material for the production of self-adhesive tapes with increased thermal resistance. It is assumed that self-adhesive materials achieving this criterion should retain their functional properties at least up to 150 °C. This value, because of the unique properties of Si-PSAs, is commonly found even for unmodified tapes; however, owing to the high cost of production, they are not widely used and are more of a material for special applications [7,8,9,10,11].

Owing to their unique properties, resulting from the structure of the molecules (no functional groups), it is not easy to modify silicone pressure-sensitive adhesives. At present, cross-linking is considered a chemical modification of ready-made silicone resins. For other chemical modifications, it is already necessary to design polymers at the stage of their synthesis. The second possibility of modification of silicone resins—currently the most used—is physical modification consisting of the introduction of plasticizers or fillers to add new ones or to strengthen the properties already possessed by Si-PSAs [12,13,14,15,16,17,18].

This work presents the physical modification of commercial silicone resin with selected clay fillers from the dellite series to obtain tapes with increased thermal resistance. Commercially purified and modified quaternary ammonium salt fillers were selected to compare the effect on the performance of Si-PSAs.

## 2. Materials and Methods

### 2.1. Materials

In the presented research, a silicone adhesive from Dow Corning (Midland, MI, USA) was used: DOWSIL ™ 7358 (Q2-7358). The cross-linking compound was dichlorobenzoyl peroxide—DClBPO (Gelest, Morrisville, PA, USA), while the solvent was toluene from Carl Roth (Karlsruhe, Germany). Dellite HPS, Dellite LVS, Dellite CW9, and Dellite 67G from Laviosa Chimica Mineraria S.P.A. (Livorno, Italy) were used as fillers.

### 2.2. Preparation of Adhesives

In the first stage of research, the starting pressure-sensitive adhesive was characterized. The number of solids was measured (thermogravimetric method) and the result was 57.6%. An initial viscosity test was also performed: 16.7 Pas. Such an adhesive was modified with a thermal cross-linking compound dibenzoyl peroxide in the amount of 2.0 pph (the result of the authors’ previous research) and with toluene to 55% solids. The resulting composition was mixed by hand and then four different fillers were added in varying amounts (from 0.1 to 3): Dellite HPS, Dellite LVS, Dellite CW9, and Dellite 67G, by Laviosa Chimica Mineraria S.P.A. Characteristics of the fillers are presented in Table 1.

### 2.3. Preparation of Self-Adhesive Tapes

To obtain a one-sided self-adhesive tape necessary for the tests, the pressure-sensitive adhesive must be coated on a polyester foil (foil weight: 50 g/m^2^). Using a semi-automatic coater, a composition with a basis weight of 45 g/m^2^ was coated and placed into an oven to cross-link and evaporate the solvent. The oven drying conditions are shown in Table 2.

### 2.4. Methods

#### 2.4.1. Characterization of the Filler

##### Fourier Transform Infrared Spectroscopy

The FT-IR spectra of MM were performed with the Thermo Nicolet 380 (Waltham, MA, USA) spectrometer with an ATR unit for wavenumbers from 400 to 4000 cm^−1^.

##### X-ray Diffraction

X-ray diffraction (XRD) analyses were performed to evaluate the modification of montmorillonite. The XRD patterns were recorded by an Empyrean PANalytical (Malvern, UK) X-ray diffractometer with the Cu lamp used as the radiation source in the 2θ range 5–60° with a step size of 0.026.

#### 2.4.2. Characterization of the Pressure-Sensitive Adhesives

The pot life of a pressure sensitive adhesive is the point at which the adhesive is too sticky to be coated. The time and the viscosity of the composition are determined [19]. As a rule, pot life is defined as the time needed to double the initial viscosity or quadruple it (for products with a lower viscosity). The measurement is carried out at room temperature and starts immediately after mixing the sample [20]. The study was performed with the use of a DV-II Pro Extra viscometer (Brookfield, New York, NY, USA). Using a moisture analyzer (Radwag MAX 60/NP, Radom, Poland), the solids content of a commercial pressure-sensitive adhesive was determined. Using a round stamp (with an area of 10 cm^2^), the basis weight of the pressure-sensitive adhesive after the cross-linking process was determined (Karl Schröder, Weinheim, Germany).

#### 2.4.3. Characterization of the Adhesive Tape

##### Peel Adhesion Testing

Peel adhesion is one of the basic phenomena that is very important in the adhesive technology. It is defined by the interaction of the surfaces of the layers of physical bodies or phases, which allows the transfer of loads between them. In other words, peel adhesion is related to the action of the forces of attraction between the phases on the contact surface of the materials. It is a very complex process and very difficult to define unequivocally [21,22]. The peel adhesion was measured using the FINAT FTM 1 method. In this method, the adhesion is tested by peeling the adhesive film off the surface at a speed of 300 mm/min at an angle of 180° [23]. The result of the test is the value of the force of peeling off the sample from the plate. An adhesive material 25 mm wide and at least 127 mm long is stuck to a steel plate and then placed in the jaws of a testing machine [24].

##### Tack

Tack is defined as the ability of an adhesive to form a bond of measurable strength to another surface under low pressure and short contact time conditions. This property can be measured using various methods: the rolling ball tack test, the quick stick tack test, the probe tack test, and the loop tack test. In the technology of obtaining pressure-sensitive adhesives, the loop adhesion test according to the AFERA 4015 standard is most often used [25,26]. According to this method, the force that is necessary to separate (at a certain speed) a tile with a certain surface and a loop, covered with adhesive, is tested. Metal, glass, and plastic plates are used. The entire test is carried out using a testing machine [27].

##### Cohesion Test

The phenomenon by which bodies resist tearing them apart is called cohesion. It is the most important property of pressure-sensitive adhesives, which determines their subsequent application because it is responsible for the cohesion and strength of the adhesive joint. The following elements affect the cohesion: the type of monomer and polymer used, the molecular weight of the adhesive, the thickness of the adhesive joint, the concentration and type of cross-linking compounds, and the temperature value during the tests [28,29]. The cohesion of the adhesive joint was determined using the FINAT FTM 8 method: shear strength from the standard surface. It consists of measuring the durability of the adhesive joint, tested at room temperature and 70 °C, under a load of 10 N. To perform the test, a strip of self-adhesive material was stuck to a steel plate. The contact surface of the adhesive and the tile was 25 mm by 25 mm. After about 10 min, the other end of the strip was loaded with a kg weight. The result of this test is the time it takes for the sample to detach from (fall off) the plate [30,31].

##### SAFT Test

SAFT is an abbreviation of shear adhesion failure temperature [32]. The SAFT test was performed in the same way as the cohesion. The only difference is raising the temperature in the furnace from 25 to 225 °C with a heating rate of 0.4 °C/min [33].

##### Shrinkage

Shrinkage is defined as the change in film dimensions at 70. Shrinkage was measured using the cross method. The test was performed on a PVC foil covered with adhesive and stuck to a metal plate. Then, two right-angle cuts were made. After the specified time, the size of the cuts was measured at eight points. The measurement result is the average of the eight measurements [34].

## 3. Results

The results for the silicone pressure-sensitive adhesive crosslinking by 1.5 wt.% crosslinking agent without filler modification are presented in Table 3. The selected composition for modification with fillers was characterized by relatively high values of adhesion and tack, very good cohesion at room and elevated temperatures, and good high thermal resistance.

To obtain new silicone pressure-sensitive adhesives’ compositions, different types of dellite were added in the preparation process. Four of the commercially available fillers were selected—where two were a nanoclay deriving from a naturally occurring, especially purified montmorillonite (Dellite HPS and Dellite LVF) and the other two were a nanoclay deriving from a naturally occurring montmorillonite, especially purified and modified with a high content of quaternary ammonium salt (Dellite CW9 and Dellite 67G). The FT-IR spectra (Figure 1) showed a typical board absorption band for clean nanoclay filler (blue and green curve)—at around 3500 and 1650 cm^−1^ OH stretching and hydration, respectively; 1030 cm^−1^ Si-O stretching, in-plane; 530 cm^−1^ Si-O-Al deformation vibration; and 450 cm^−1^ Si-O-Si bending [35,36]. The Dellite 67 G and Dellite CW9 spectra are characterized by a significant decrease in the relative intensity of the bands at around 3500 and 1650 cm^−1^ compared with Dellite HPS and Dellite LVF, which indicated that the surface of modified dellite changed from a hydrophilic to a hydrophobic character and the content of water decreased [35,37]. Moreover, for Delite CW9, absorption bonds were observed, which is typical for ammonium salt modification and proves the high hydrophobicity of the material [38].

X-ray diffraction was used to confirm the nanoclay (montmorillonite) characteristics’ patterns (green and blue curves) and modificated dellite by ammonium salt fillers. The modification was observed in Figure 2—on the graph, an increase in d-spacing is visible for modified dellite (red and black curves) and the shift is much more evident, which might suggest the use of more modifying compound [38,39].

The Figure 3 shows the peel adhesion results for silicone pressure-sensitive adhesives composition modified with different types of dellite. The addition of dellite 67 G and dellite LVF causes initial increased/stabilization and then a slightly decreased the value of adhesion. The lowest degree of filling dellite HPS obtained the highest adhesion value—about 15 N/25 mm. Only its highest tested degree of filling caused a decrease in adhesion to the vicinity of the initial value of the sample not modified with the filler (about 13 N/25 mm). Only for dellite LVF does addition of a filler cause initial increased and back to the basic level of adhesion value. The dellite CW9 caused a slow reduction in the adhesion value. Each modification had a different effect on the properties of the adhesive tape. Both the initial increase and the decrease in the value of adhesion could be caused by the compatibility of the filler in the matrix, its miscibility, and the tendency to agglomerate [26,34]. The most typical behavior in the matrix was characterized as dellite CW9.

The results of additions of filler on tack are collected in Figure 4. The high compatibility of all fillers and the matrix caused an increase in the tack value for the lowest content of filler; then, the value of tack decreased, which is characteristic for the increased filler agglomeration. It is well known that this effect is clearly visible at low and high polymer matrix fillings [14,26,34].

Figure 5 and Figure 6 show the cohesion of the tapes obtained at room temperature and elevated temperature. For the temperature of 20 °C in all tested compositions, values higher than the required values for the tapes produced in the industry (above 72 h) were observed. In the case of cohesion determined at 70 °C, lower results were not observed. Very good results were obtained with the modified compositions. It follows that the addition of all types of dellite does not cause a decrease in cohesion.

To determine the maximum operating temperature, the SAFT test of self-adhesives tape was performed (Figure 7). All modifications increased the thermal resistance of the obtained self-adhesive tapes. The addition of a small amount of filler caused a drastic increase in the thermal resistance of the material and thermal resistance with the addition of 0.1% pph. It increased from 150 to 220 °C or 225 °C (the maximum of the study), i.e., it increased by 70–75 degrees. With an increase in the filler content on dellite-modified ammonium salt, a decrease in thermal resistance was noted and, at 3 wt., a drastic decrease (below the thermal resistance of unmodified glue felts), which may result from the modification of ammonium salt, which increases the tendency of the filler to agglomerate, and thus causes uneven distribution in the adhesive tape [15,16,18,38]. However, for unmodified fillers, no significant changes were noted depending on the degree of filling.

The influence of different amounts of dellite fillers on shrinkage silicone pressure-sensitive adhesives film is collected in Table 4 and Table 5, presented in numerical and visual form. For comparison, the shrinkage over time of the composition without the filler is also shown. For each of the fillers used, the smaller the shrinkage, the higher the filler content in the polymer matrix. This is because of better alignment of the polymer mesh and a more compact internal structure of the adhesive film [17,26]. Moreover, it turned out that the addition of a filler not modified with ammonium salts has a more beneficial effect on the peel of the adhesive film. This is most likely because of the fact that the introduced modifying material influences the content of silicon material in measured samples (it significantly reduces it); therefore, compared with samples modified with unmodified material, they showed lower shrinkage [18,38]. When comparing the results of unmodified fillers, no significant changes in shrinkage were noted; the same was true when comparing the influence of modified fillers.

In Figure 8, the influence of the addition of different types of dellite on viscosity over time is presented. The change in viscosity is marked in black for the adhesives without filler. For all samples, the viscosity increases with the increasing test time. For the compositions containing nanoclay, the viscosity is lower compared with the base adhesive. For each tested filler, the lowest viscosity values were obtained for the lowest fillings. The higher the degree of filling, the higher the viscosity of the composition, although the viscosities of the filled compositions were lower than the original composition. The highest increase in viscosity with all compositions is seen between the fifth and seventh days. In each case, the compositions after 7 days obtained a viscosity value that made their application impossible.

## 4. Conclusions

As a result of the research, innovative materials (one-sided self-adhesives tapes based on silicone pressure-sensitive adhesives) exhibiting increasing thermal resistance were obtained. The new materials maintained useful properties (self-adhesives properties) of adhesion, cohesion, and tack, while increasing the thermal resistance from 150 to 220–225 °C thanks to the modification of the visual adhesive composition, consisting of the introduction of silicon fillers into the resin, and their good dispersion.

From an economic point of view, a great advantage is the possibility of obtaining materials with maximum utility values using small amounts of filler. This allows to exclude the risk of filler agglomeration and the relatively simple physical modification makes it easy to apply on existing production lines.

The materials obtained can be successfully used in heat engineering, covering installations and fireplaces connecting elements exposed to high temperature in households; in hot-stamping technology; in the heavy industry; and in the automotive industry, as tapes connecting materials operating at elevated temperatures, as well as in aeronautics as a binder for solar batteries on satellite decks, masking tapes in powder coating.

## Figures and Tables

**Figure 1 materials-15-07460-f001:**
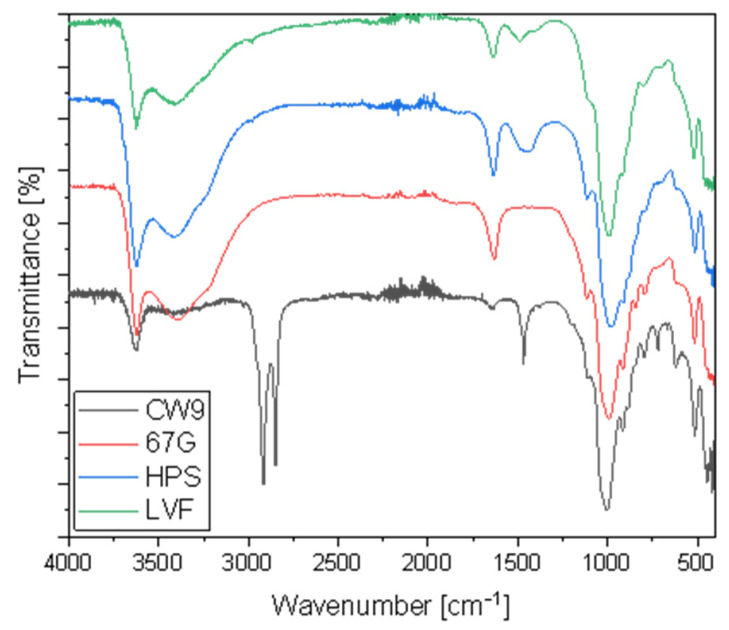
Fourier transform infrared (FTIR) spectra for different types of nanoclay.

**Figure 2 materials-15-07460-f002:**
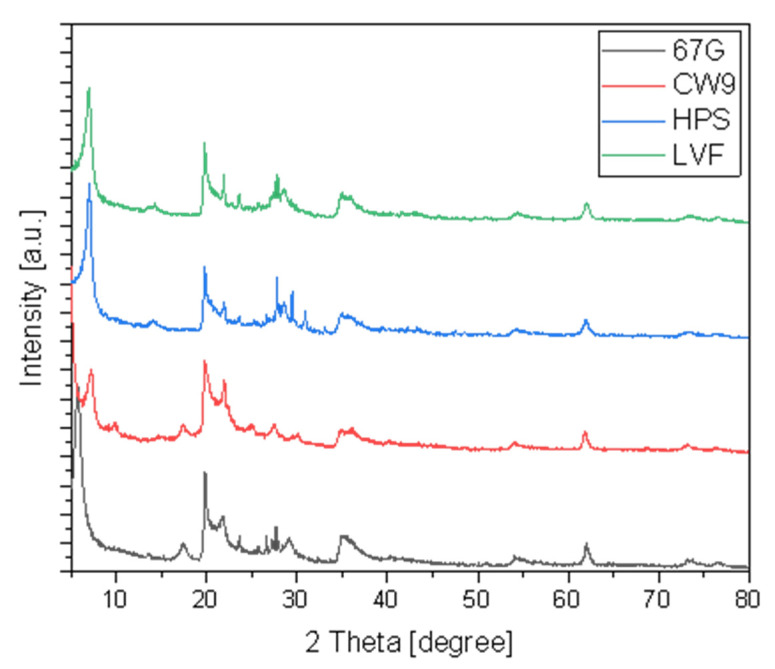
XRD patterns of different types of nanoclay.

**Figure 3 materials-15-07460-f003:**
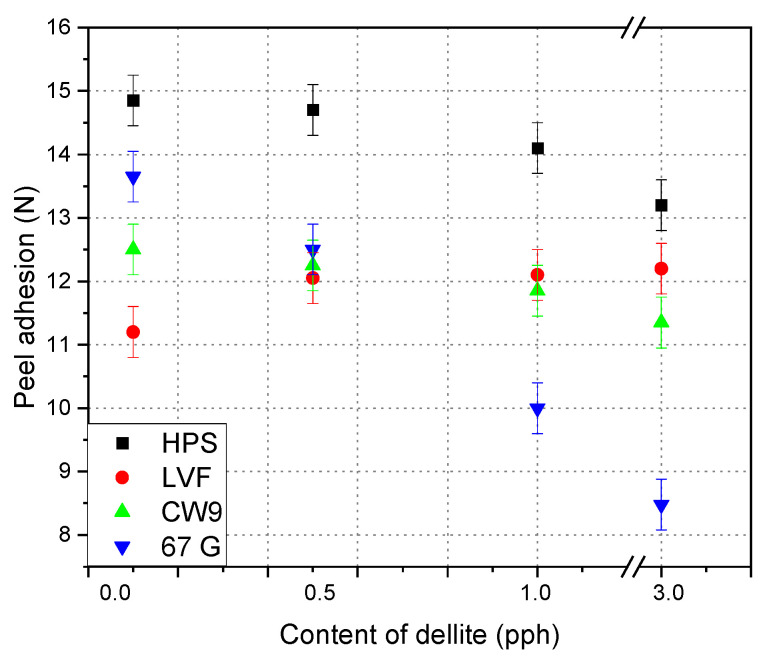
Effect of contents of dellite addition on the peel adhesion of silicone pressure-sensitive adhesives.

**Figure 4 materials-15-07460-f004:**
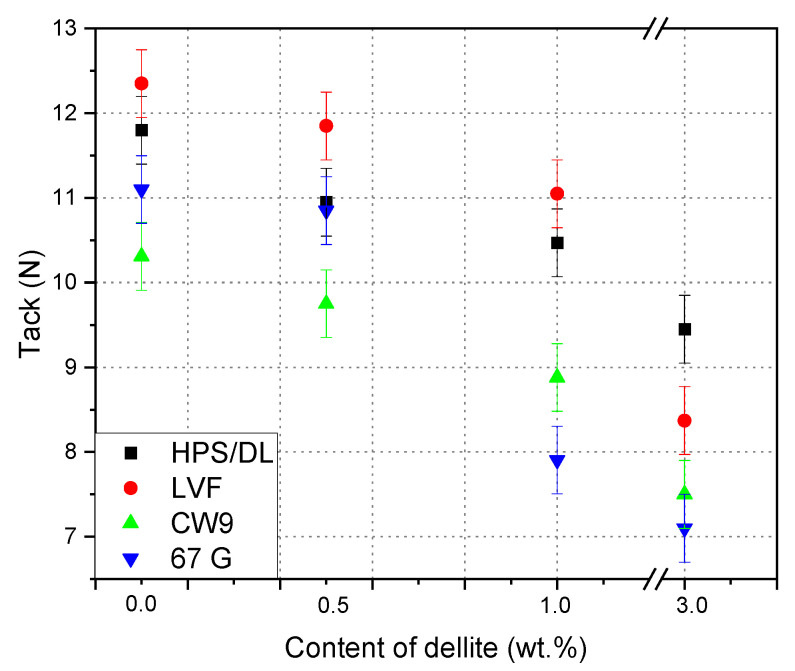
Effect of contents of dellite addition on the tack of silicone pressure-sensitive adhesives.

**Figure 5 materials-15-07460-f005:**
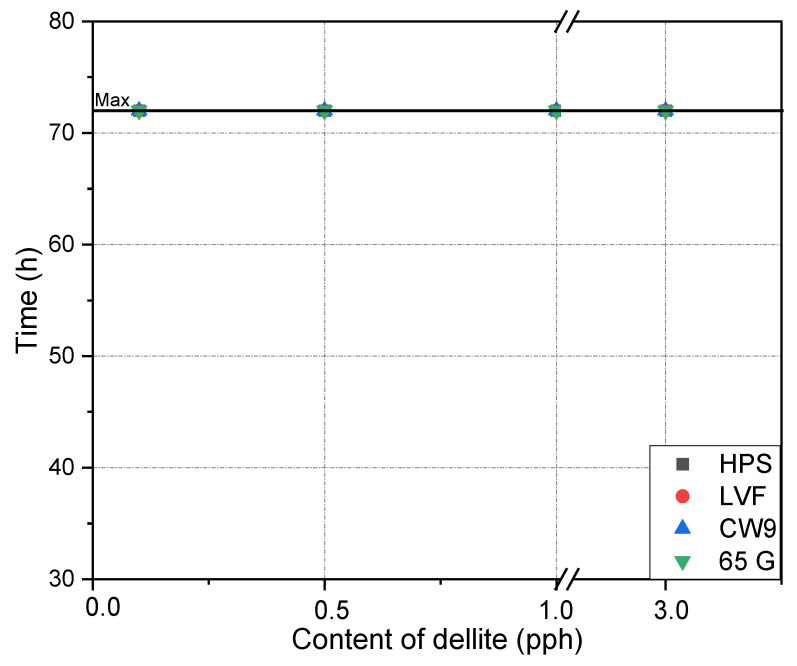
Effect of contents of dellite addition on the cohesion at 20 °C of silicone pressure-sensitive adhesives.

**Figure 6 materials-15-07460-f006:**
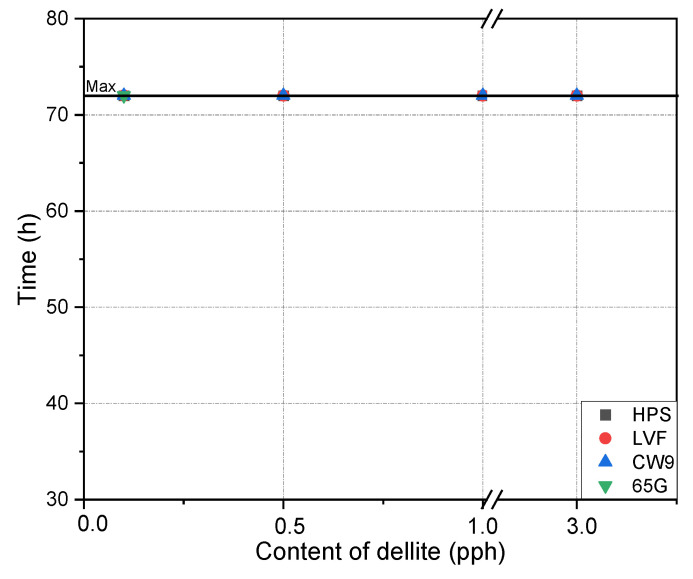
Effect of contents of dellite addition on the cohesion at 70 °C of silicone pressure-sensitive adhesives.

**Figure 7 materials-15-07460-f007:**
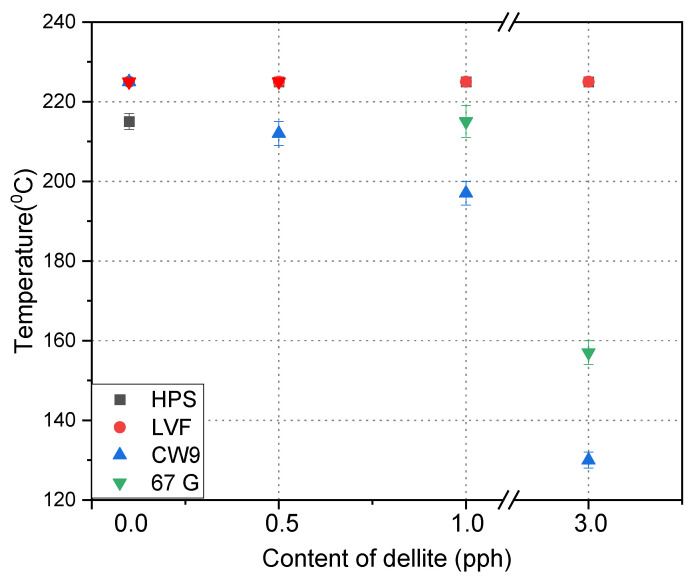
Effect of contents of dellite addition on the maximum work temperature of silicone pressure-sensitive adhesives.

**Figure 8 materials-15-07460-f008:**
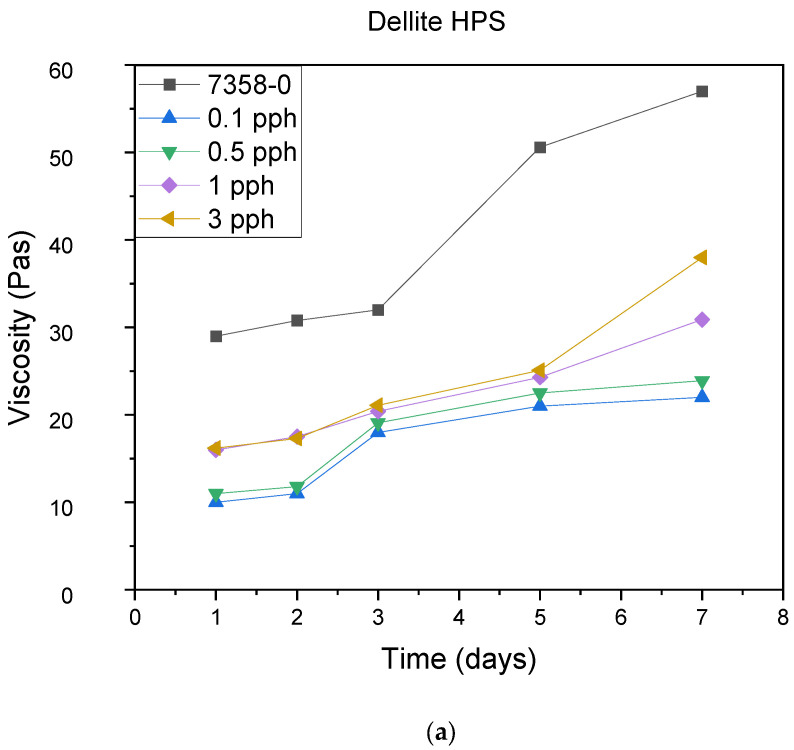
Pot life of pressure-sensitive adhesives: marked with 0—adhesive without modification. (**a**) Dellite HPS, (**b**) Dellite LVF, (**c**) Dellite CW9, and (**d**) Dellite 67G.

**Table 1 materials-15-07460-t001:** Chemical and physical data of the tested fillers.

	Dellite HPS	Dellite LVF	Dellite CW9	Dellite 67G
Description	a nanoclay deriving from a naturally occurring, especially purified montmorillonite.		a nanoclay deriving from a naturally occurring montmorillonite, especially purified and modified with a high content of quaternary ammonium salt (dimethyl dihydrogenated tallow ammonium)	
Color	Whitish	White	Off white	
Moisture [%]	4–8		3	
Loss of ignition [weight%]	4–6		43–48	40–45
Particle size (dry) µm]	7–9		15–20	7–9
Modifier	-		Dimethyl dehydrogenated tallow ammonium	
Specific weight [g/cm^3^]	2.2		1.7	
Bulk density [g/cm^3^]	0.65		0.45	
Cation exchange Capacity [Meq/100 g]	128	105	-	-

**Table 2 materials-15-07460-t002:** The conditions of preparation of adhesives and adhesive films.

Name of Filler	Content of Filler (pph)	ES ^1^: Time (min)/Temperature (°C)	Coat Weight ^2^ (g/m^2^)
Dellite HPS	0.1	10/110	45
	0.5		
	1		
	3		
Dellite LVS	0.1	10/110	45
	0.5		
	1		
	3		
Dellite CW9	0.1	10/110	45
	0.5		
	1		
	3		
Dellite 67G	0.1	10/110	45
	0.5		
	1		
	3		

^1^ ES, evaporation of solvent; ^2^ coat weight of adhesive film.

**Table 3 materials-15-07460-t003:** Q2 7358 pressure-sensitive adhesive results without modification.

Peel Adhesion [N/25 mm]	Cohesion [h]		SAFT (°C)	Tack [N]
	20 °C	70 °C		
**12.75**	**>72**	>72	150	10.45

**Table 4 materials-15-07460-t004:** Shrinkage of pressure-sensitive adhesives: marked with 0—adhesive without modification. (a) Dellite HPS, (b) Dellite LVF, (c) Dellite CW9, and (d) Dellite 67G.

(a)												
Shrinkage (%)												
Content of Additives (pph)	10 min	30 min	1 h	3 h	8 h	24 h	2 Days	3 Days	4 Days	5 Days	6 Days	7 Days
**0 ***	**0.42**	**0.41**	**0.64**	**0.90**	**0.96**	**1.02**	**1.10**	**1.14**	**1.33**	**1.33**	**1.33**	**1.33**
0.1	0.129 ±0.005	0.201 ±0.003	0.241 ±0.005	0.262 ±0.002	0.281 ±0.003	0.301 ±0.004	0.315 ±0.003	0.333 ±0.005	0.352 ±0.005	0.391 ±0.003	0.425 ±0.004	0.496 ±0.004
0.5	0.121 ±0.004	0.187 ±0.005	0.229 ±0.005	0.254 ±0.005	0.272 ±0.003	0.289 ±0.005	0.314 ±0.005	0.328 ±0.002	0.352 ±0.005	0.382 ±0.005	0.442 ±0.003	0.475 ±0.003
1.0	0.097 ±0.003	0.154 ±0.004	0.200 ±0.005	0.235 ±0.003	0.258 ±0.002	0.279 ±0.003	0.299 ±0.005	0.313 ±0.005	0.351 ±0.004	0.410 ±0.003	0.441 ±0.005	0.446 ±0.002
3.0	0.071 ±0.005	0.121 ±0.002	0.134 ±0.005	0.154 ±0.002	0.174 ±0.003	0.201 ±0.005	0.229 ±0.005	0.255 ±0.005	0.298 ±0.003	0.345 ±0.004	0.412 ±0.005	0.412 ±0.004
(**b**)												
**Shrinkage (%)**												
**Content of Additives (pph)**	**10 min**	**30 min**	**1 h**	**3 h**	**8 h**	**24 h**	**2 Days**	**3 Days**	**4 Days**	**5 Days**	**6 Days**	**7 Days**
0.1	0.134 ±0.004	0.210 ±0.005	0.292 ±0.003	0.321 ±0.002	0.346 ±0.003	0.355 ±0.004	0.371 ±0.002	0.383 ±0.003	0.432 ±0.005	0.464 ±0.004	0.501 ±0.004	0.533 ±0.003
0.5	0.126 ±0.003	0.154 ±0.002	0.205 ±0.004	0.235 ±0.003	0.274 ±0.005	0.310 ±0.005	0.354 ±0.005	0.370 ±0.005	0.397 ±0.004	0.454 ±0.003	0.478 ±0.005	0.525 ±0.005
1.0	0.087 ±0.004	0.152 ±0.005	0.197 ±0.003	0.244 ±0.004	0.318 ±0.005	0.342 ±0.003	0.358 ±0.003	0.391 ±0.003	0.402 ±0.005	0.432 ±0.002	0.454 ±0.003	0.497 ±0.004
3.0	0.092 ±0.003	0.212 ±0.005	0.179 ±0.002	0.281 ±0.003	0.301 ±0.005	0.324 ±0.004	0.345 ±0.002	0.368 ±0.003	0.359 ±0.005	0.389 ±0.004	0.454 ±0.003	0.454 ±0.005
(**c**)												
**Shrinkage (%)**												
**Content of Additives (pph)**	**10 min**	**30 min**	**1 h**	**3 h**	**8 h**	**24 h**	**2 Days**	**3 Days**	**4 Days**	**5 Days**	**6 Days**	**7 Days**
0.1	0.221 ±0.004	0.314 ±0.005	0.396 ±0.003	0.432 ±0.004	0.454 ±0.005	0.513 ±0.002	0.545 ±0.003	0.596 ±0.004	0.610 ±0.005	0.645 ±0.003	0.649 ±0.005	0.651 ±0.004
0.5	0.198 ±0.005	0.284 ±0.003	0.362 ±0.005	0.391 ±0.002	0.421 ±0.003	0.451 ±0.004	0.472 ±0.005	0.496 ±0.005	0.512 ±0.003	0.531 ±0.005	0.549 ±0.004	0.567 ±0.005
1.0	0.171 ±0.003	0.144 ±0.005	0.347 ±0.003	0.174 ±0.005	0.238 ±0.002	0.341 ±0.003	0.357 ±0.005	0.488 ±0.003	0.387 ±0.004	0.406 ±0.002	0.425 ±0.002	0.554 ±0.003
3.0	0.116 ±0.004	0.098 ±0.002	0.325 ±0.003	0.248 ±0.004	0.303 ±0.004	0.340 ±0.005	0.366 ±0.005	0.454 ±0.002	0.389 ±0.003	0.436 ±0.003	0.546 ±0.004	0.546 ±0.005
(**d**)												
**Shrinkage (%)**												
**Content of Additives (pph)**	**10 min**	**30 min**	**1 h**	**3 h**	**8 h**	**24 h**	**2 Days**	**3 Days**	**4 Days**	**5 Days**	**6 Days**	**7 Days**
0.1	0.263 ±0.004	0.314 ±0.003	0.354 ±0.004	0.379 ±0.005	0.397 ±0.004	0.425 ±0.003	0.461 ±0.002	0.496 ±0.004	0.514 ±0.003	0.526 ±0.003	0.543 ±0.004	0.559 ±0.005
0.5	0.189 ±0.002	0.289 ±0.005	0.347 ±0.003	0.356 ±0.004	0.389 ±0.003	0.421 ±0.005	0.455 ±0.005	0.488 ±0.005	0.499 ±0.002	0.512 ±0.003	0.521 ±0.005	0.533 ±0.003
1.0	0.163 ±0.004	0.257 ±0.005	0.339 ±0.004	0.357 ±0.005	0.373 ±0.002	0.401 ±0.002	0.415 ±0.005	0.428 ±0.003	0.446 ±0.005	0.489 ±0.004	0.514 ±0.005	0.520 ±0.003
3.0	0.105 ±0.003	0.242 ±0.002	0.312 ±0.005	0.342 ±0.003	0.359 ±0.005	0.399 ±0.004	0.403 ±0.003	0.417 ±0.005	0.432 ±0.004	0.456 ±0.003	0.504 ±0.004	0.504 ±0.004

* 0—adhesive without modification.

**Table 5 materials-15-07460-t005:** Shrinkage of pressure-sensitive adhesives: marked with 0—adhesive without modification.

Name		10 Min	1 h	3 Days	7 Days
**Dellite HPS/DL**	**0.1**	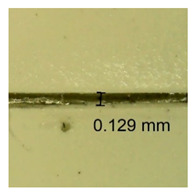	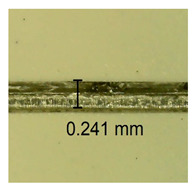	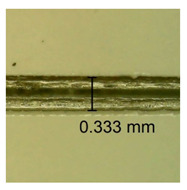	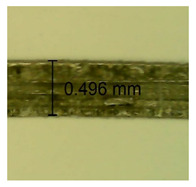
	**0.5**		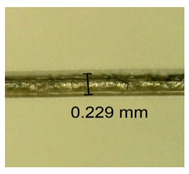	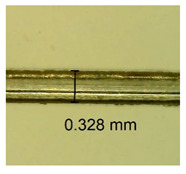	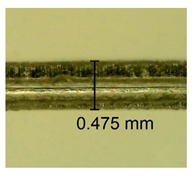
	**1.0**		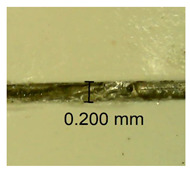		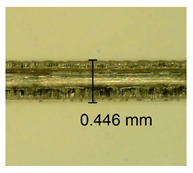
	**3.0**	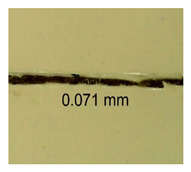	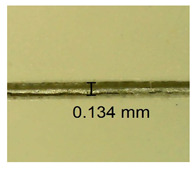	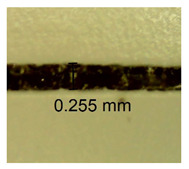	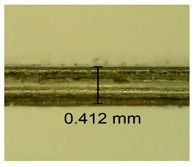
**Dellite LVF**	**0.1**	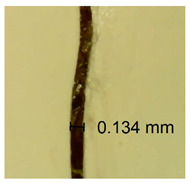	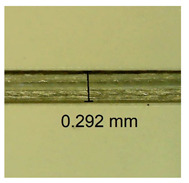	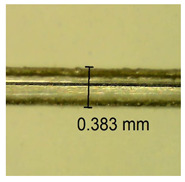	
	**0.5**	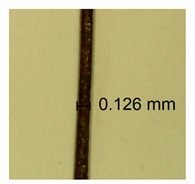	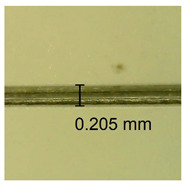	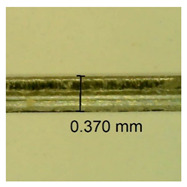	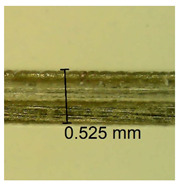
	**1.0**	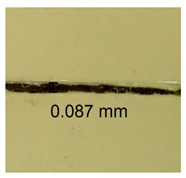	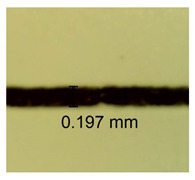	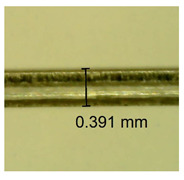	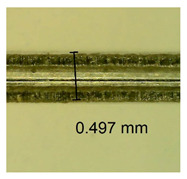
	**3.0**	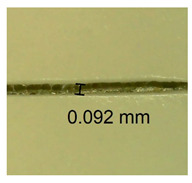	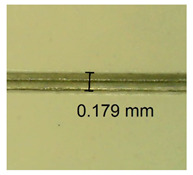	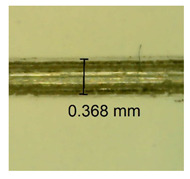	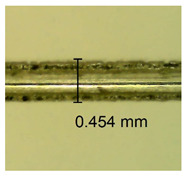
**Dellite CW9**	**0.1**	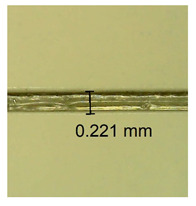	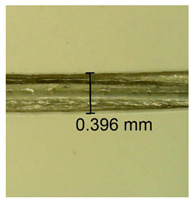	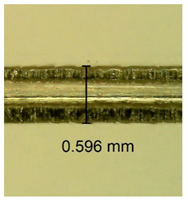	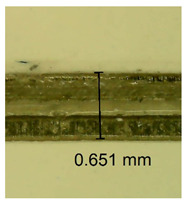
	**0.5**	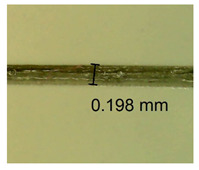	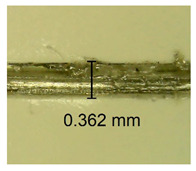	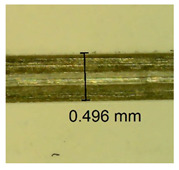	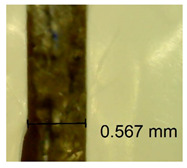
	**1.0**	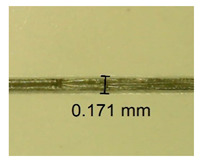	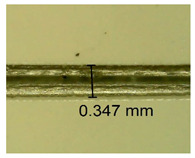	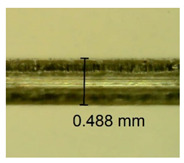	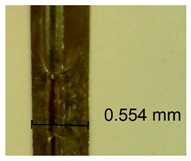
	**3.0**	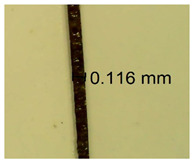	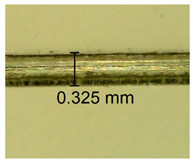	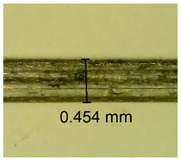	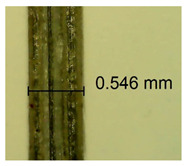
**Dellite 67 G**	**0.1**	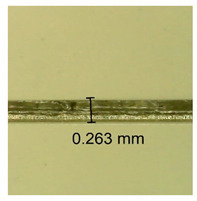	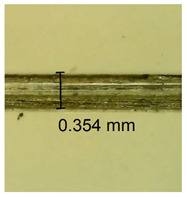	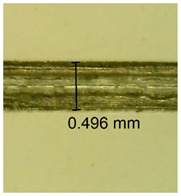	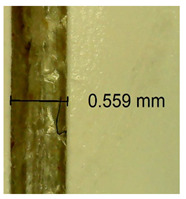
	**0.5**	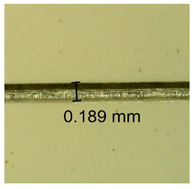	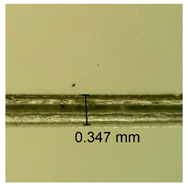	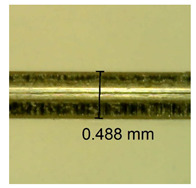	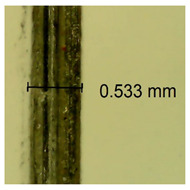
	**1.0**	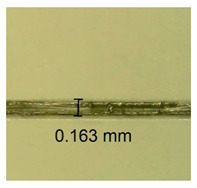	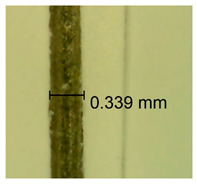	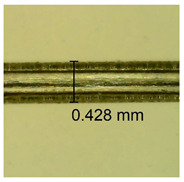	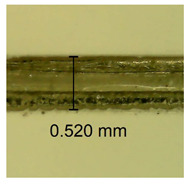
	**3.0**		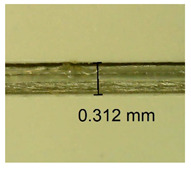	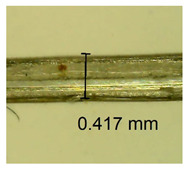	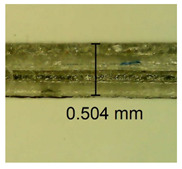

## Data Availability

Not applicable.

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
