# Peer review of "Influence of Nanoclay on the Thermo-Mechanical Properties of Silicone Pressure-Sensitive Adhesives"

_materials, 2022, doi:10.3390/ma15217460_

Round 1

Reviewer 1 Report

The paper by Antosik and Mozelewska examines the addition of nano clays, and their effects, on silicone pressure-sensitive adhesives. This is a generally well written manuscript, and I found it easy to read. However, there are some minor issues that need to be corrected prior to publication.

There are inconsistencies in the capitalization of proper names (Delitte vs delitte), Figure vs figure, and others throughout the manuscript. Table 2 could be removed as all of the provided information is the same for all fillers; this information could be combined into a single sentence in the text. Figure captions could be more descriptive and any abbreviations used for the different types of fillers need to be explicitly stated. Figures 5 and 6 could be combined into a two-panel figure as the data is fairly monotonous. There is no in text reference to Figure 7. The data in Tables 3 and 4 could be moved to a Supporting Information section, I don't need to see everything, especially if only a passing reference is made to the data/images. The referencing style needs to be checked for form; there are many inconsistencies.

Some further suggestions:

Line 4 - Remove the comma in the author name

Line 31-36 - Do all of these different clay minerals need to be listed? How do they contribute to the manuscript as a whole?

Line 75 - Change "The" to "This"

Line 107 - Please define the abbreviation "MM"

Line 116-117 - This sentence seems to suggest that the adhesive is being coated by something but should imply imply that that the adhesive is too sticky to be used as a coated. Please correct the grammar.

Line 117 - The sentence beginning with "The time..." is confusing. Provide an explanation as to how the time of the composition was determined, or correct the grammar.

Line 160 - Include a value with kg...is this intended to be read was 1 kg?

Line 167 - Include units with the number 70...70 °C??

Line 179 - Change obtained to "obtain".

Line 180 - Change addition to "added".

Line 197 - Change "type" to "types"

Line 200 - "graphs" should be replaced with XRD patterns...or something more appropriate.

Line 204 - Change "type" to "types"

Line 210 - Correct "ini-tial"

Line 211 - This sentence is grammatically incorrect.

Line 220 - This sentence is confusing as written.

Line 245 - "3 wt." If this is intended to communicate a w/w% please correct.

Line 257 - Capitalize "This".

Line 272 - Correct figures to "figure".

Author Response

The authors would like to thank to the Reviewer and truly appreciate his comments, questions and corrections. Please find the detailed answers below.

The paper by Antosik and Mozelewska examines the addition of nano clays, and their effects, on silicone pressure-sensitive adhesives. This is a generally well written manuscript, and I found it easy to read. However, there are some minor issues that need to be corrected prior to publication.

There are inconsistencies in the capitalization of proper names (Delitte vs delitte), Figure vs figure, and others throughout the manuscript. Table 2 could be removed as all of the provided information is the same for all fillers; this information could be combined into a single sentence in the text. Figure captions could be more descriptive and any abbreviations used for the different types of fillers need to be explicitly stated. Figures 5 and 6 could be combined into a two-panel figure as the data is fairly monotonous. There is no in text reference to Figure 7. The data in Tables 3 and 4 could be moved to a Supporting Information section, I don't need to see everything, especially if only a passing reference is made to the data/images. The referencing style needs to be checked for form; there are many inconsistencies.

Thank you for drawing our attention to the above suggestions. We tried to make the text more consistent, broaden the descriptions and adapt to the above-mentioned comments.

Some further suggestions:

Line 4 - Remove the comma in the author name

We have corrected the manuscript.

Line 31-36 - Do all of these different clay minerals need to be listed? How do they contribute to the manuscript as a whole?

We have corrected the manuscript.

Line 75 - Change "The" to "This"

We have corrected the manuscript.

Line 107 - Please define the abbreviation "MM"

We have corrected the manuscript.

Line 116-117 - This sentence seems to suggest that the adhesive is being coated by something but should imply imply that that the adhesive is too sticky to be used as a coated. Please correct the grammar.

We made a change to the manuscript, we hope now it is correct.

Line 117 - The sentence beginning with "The time..." is confusing. Provide an explanation as to how the time of the composition was determined, or correct the grammar.

We made a change to the manuscript, we hope now it is correct.

Line 160 - Include a value with kg...is this intended to be read was 1 kg?

We have corrected the manuscript.

Line 167 - Include units with the number 70...70 °C??

We have corrected the manuscript.

Line 179 - Change obtained to "obtain".

We have corrected the manuscript.

Line 180 - Change addition to "added".

We have corrected the manuscript.

Line 197 - Change "type" to "types"

We have corrected the manuscript.

Line 200 - "graphs" should be replaced with XRD patterns...or something more appropriate.

We made a change to the manuscript, we hope now it is correct.  

Line 204 - Change "type" to "types"

We have corrected the manuscript.

Line 210 - Correct "ini-tial"

We have corrected the manuscript.

Line 211 - This sentence is grammatically incorrect.

We have corrected the manuscript.

Line 220 - This sentence is confusing as written.

We have corrected the manuscript.

Line 245 - "3 wt." If this is intended to communicate a w/w% please correct.

We have corrected the manuscript to pph. 

Line 257 - Capitalize "This".

We have corrected the manuscript.

Line 272 - Correct figures to "figure".

We have corrected the manuscript.

Finally, we hope that corrections made in the manuscript fulfill reviewer suggestions and allow editor to make positive decision about acceptation of our contribution for publishing in this journal.

With regards,

Adrian Krzysztof Antosik

Karolina Mozelewska

Reviewer 2 Report

In the review of research article titled: Influence of nanoclay on the thermo-mechanical properties of silicone pressure-sensitive adhesives, the authors have described the research work very well covering a lot of aspects. I would like to see this article publish but after some minor modifications as follow;

1.      I would like to suggest the authors to highlight (values) of their achieved results in abstract clearly rather than focusing on general discussion.

2.      Problem statement in the introduction portion is missing, what were the drawbacks in previous study, which have compelled the authors to carry out this study and specifically this material?

3.      In the FTIR results, I would like to suggest the authors to index the bands within the Figure to make it more attractive.

4.      Please index the XRD peaks according to structure and space group. Please mention the PDF card number for indexing of peaks.

5.      I would suggest the authors to calculate the crytallinity by the XRD analysis as well to make this work more appealing.

6.      There exist several typing errors and grammatical mistakes, please revise the manuscript carefully by some native English speaker.

Author Response

The authors would like to thank to the Reviewer and truly appreciate his comments, questions and corrections. Please find the detailed answers below.

In the review of research article titled: Influence of nanoclay on the thermo-mechanical properties of silicone pressure-sensitive adhesives, the authors have described the research work very well covering a lot of aspects. I would like to see this article publish but after some minor modifications as follow;

  1. I would like to suggest the authors to highlight (values) of their achieved results in abstract clearly rather than focusing on general discussion.

We made a change to the manuscript, we hope now it is correct.

  1. Problem statement in the introduction portion is missing, what were the drawbacks in previous study, which have compelled the authors to carry out this study and specifically this material?

We made a change to the manuscript, we hope now it is correct.

  1. In the FTIR results, I would like to suggest the authors to index the bands within the Figure to make it more attractive.

We made a change to the manuscript, we hope now it is correct. 

  1. Please index the XRD peaks according to structure and space group. Please mention the PDF card number for indexing of peaks.

Our article is about adhesives and self-adhesive tapes modified with commercial fillers. Unfortunately, in our research we did not make the fillers ourselves, nor did we modify them with other compounds. We used X-ray diffraction only to confirm the safety data sheet of the purchased materials. Dellite in two cases was modified with ammonium salt, but this modification was made by the manufacturer. Therefore, we felt that it was not necessary to determine the structure and functional groups of the fillers. We performed the XRD test and added it to the article only to verify and confirm that we have received the correct fillers. Nevertheless, if we make any modifications, or even confirm the characteristics of the fillers in subsequent articles, we will perform such an analysis and add it to the article. Thank you very much for the above remark and by writing another article from our research, we will make such an analysis.

  1. I would suggest the authors to calculate the crytallinity by the XRD analysis as well to make this work more appealing.

Thank you very much for the comment presented. As we wrote above, the XRD study was to serve as a confirmation of the structure of the compounds purchased. Moreover, in our research we used a maximum of 3 pph of given fillers. This is a very small amount in relation to the weight of the entire composition. The degree of crystallinity is important in the case of higher amounts of filler in the adhesive composition. In our case, it will have little impact. Nevertheless, when we modify the adhesive composition with more filler, we will certainly use this suggestion. However, we hope that this will not affect the positive reception of our article and the corrections made by us.

  1. There exist several typing errors and grammatical mistakes, please revise the manuscript carefully by some native English speaker.

We made a change to the manuscript, we hope now it is correct. 

Finally, we hope that corrections made in the manuscript fulfill reviewer suggestions and allow editor to make positive decision about acceptation of our contribution for publishing in this journal.

With regards,

Adrian Krzysztof Antosik

Karolina Mozelewska